# Perceived Stress and Coping Behavior of Nurses Caring for Critical Patients with COVID-19 Outbreak in Taiwan: A Mixed-Methods Study

**DOI:** 10.3390/ijerph19074258

**Published:** 2022-04-02

**Authors:** Shu-Yen Lee, Kai-Jo Chiang, Yi-Jiun Tsai, Chi-Kang Lin, Yun-Ju Wang, Chou-Ping Chiou, Hsueh-Hsing Pan

**Affiliations:** 1School of Nursing, National Defense Medical Center, Taipei 11420, Taiwan; leey1108@mail.ndmctsgh.edu.tw (S.-Y.L.); carol428@mail.ndmctsgh.edu.tw (K.-J.C.); 609209004@mail.ndmctsgh.edu.tw (Y.-J.T.); twins_hcwjoann@mail.ndmctsgh.edu.tw (Y.-J.W.); 2Department of Nursing, Tri-Service General Hospital, Taipei 11420, Taiwan; 3Department of Nursing, Tri-Service General Hospital Songshan Branch, Taipei 10530, Taiwan; 4Department of Gynecology and Obstetrics, Tri-Service General Hospital, National Defense Medical Center, Taipei 11420, Taiwan; kung568@mail.ndmctsgh.edu.tw; 5School of Nursing, I-Shou University, Kaohsiung 84020, Taiwan

**Keywords:** perceived stress, coping behavior, critical patients, COVID-19

## Abstract

Severe pneumonia with novel pathogens, also called COVID-19, caused a pandemic in Taiwan as well as in the rest of the world in May 2021. Nurses are under great stress when caring for critically ill patients with COVID-19. This study aimed to explore the perceived stress and coping behaviors of nurses caring for critically ill patients with COVID-19 using a mixed-methods approach. We recruited 85 nurses from a special intensive care unit (ICU) of a medical center in Taiwan between May and June 2021. To gather data, we used a questionnaire on basic characteristics, the perceived stress scale (PSS-14), and the brief coping orientation to problems experienced inventory (B-COPE), then conducted a qualitative interview. The results showed that the average perceived stress level among nurses was 25.4 points, and most of them perceived moderate stress. The top three coping behaviors practiced by the nurses were active coping, planning, and acceptance. Nurses who received less perceived support from their friends or families and who had shorter working experience in nursing had significantly higher stress levels. The qualitative results revealed that the nurses’ perceived stress came from fear, worry, and the increased burden caused by caring for critical patients with COVID-19. Coping behaviors included rest, seeking support, and affirmative fighting. Based on these findings, it is suggested that the support nurses receive from their families is an important predictor of perceived stress. Therefore, it is suggested that nurses be provided with more support in dealing with stress caused by caring for critical patients with COVID-19 in special ICUs.

## 1. Introduction

The global COVID-19 pandemic, caused by severe acute respiratory syndrome coronavirus 2 (SARS-CoV-2), has been ongoing since December 2019. As of 11 September 2021, more than 223 million cases and 4.61 million deaths had been confirmed. The number of confirmed infections and deaths worldwide continues to increase rapidly [1]. In Taiwan, the number of confirmed infections increased daily, especially from May to June. Although the infection rate was lower than the global rate, the death rate of COVID-19 patients was 4.2%, which was higher than the global rate [2]. Faced with the impact of the COVID-19 outbreak, Taiwan’s medical environment has undergone tremendous changes, such as the establishment of screening stations and dedicated care units, which has led to a sharp increase in the psychological stress borne by nursing staff. In addition, in the early days of the COVID-19 pandemic, international news reported that many nursing staff were infected and died because of caring for patients with COVID-19. 

The high incidence of COVID-19 and the mutation of the virus may increase the stress of the first-line nursing staff when they perform nursing care [3,4]. The COVID-19 pandemic has changed the original work habits and routines of nurses. Moreover, nurses are required to work while wearing uncomfortable personal protective equipment (PPE) to prevent the risk of infection from COVID-19 patients. These factors, and especially caring for patients with COVID-19 in special intensive care units (ICUs), are sources of stress for nursing staff [5]. A prior study has shown that PPE clothing and the caregiver burden produced stress in nursing staff caring for patients with COVID-19 [6]. Many research findings also indicated that the causes of stress in nurses caring for COVID-19 patients included the burden of clinical work, social isolation, fear of contracting COVID-19, fear of infecting relatives and friends, discomfort caused by PPE, frequent hand washing and use of disinfectants, rough and damaged palms of hands, inconvenience of using the toilet at work, and restricted eating and drinking at work [5,6,7]. Among these, the greater stressors were the fear of infecting relatives and friends, uncertainty about when the pandemic will end, and seeing patients die [8]. Prior studies have identified factors that affect and increase the psychological stress of nursing staff; they include an increase in number of COVID-19 patients, increased workload, insufficient manpower, limited PPE, rapid spread of COVID-19, lack of specific treatment drugs, and lack of support [8,9].

Coping behavior refers to an individual’s response to stressors. Previous studies have indicated that while caring for severely ill COVID-19 patients, nursing staff often used coping behaviors such as taking preventive interventions, actively learning about COVID-19, adjusting their own mental perspective, actively facing the COVID-19 epidemic, seeking family support, sharing jokes with colleagues, and engaging more in humor and friendship. Through such behaviors, they tried to motivate themselves to face the pandemic with a positive attitude so as to effectively reduce stress [7,8]. The most commonly used coping strategies by nursing staff are active coping, reading about prevention, spreading information about COVID-19, and following the appropriate PPE protocol [10]. Some nurses also practice motivational self-talk to cultivate a positive perspective while dealing with the pandemic [8]. Prior studies have indicated that coping behavior is correlated with psychological well-being, and poor psychological well-being usually involves coping behaviors such as self-distraction, behavioral disengagement, denial, and venting [11]. In addition, some factors such as family support, family function, psychological status, and resilience affect coping behavior [12,13]. Previous studies have explored the experience of nurses caring for patients with COVID-19 using a qualitative research design [14] and the stress level of nurses caring for patients with COVID-19 using a cross-sectional design [8]. However, these studies did not work in ICUs, or focus on nurses. This explains why there are limited studies exploring the perceived stress and coping behavior of critical care nurses caring for COVID-19 patients in special ICUs. Therefore, this study used a mixed method design to explore the perceived stress and coping behavior of critical care nurses caring for patients with COVID-19 in a special ICU. 

## 2. Methods

### 2.1. Study Design and Participants

This was a mixed-method study design. A combination of quantitative and qualitative data was analyzed separately but integrated and discussed. Nurses were recruited from one special ICU of a medical center in northern Taiwan, which was set up expeditiously to care for critical patients with COVID-19 from May to July 2021. The inclusion criteria were as follows: (1) nurses in this special ICU who cared for critically ill patients with COVID-19, (2) nurses who understood the study purpose, and (3) nurses who agreed to participate in this study and signed the consent form. The exclusion criteria were as follows: (1) nurses who did not work in this special ICU, (2) nurses who did not directly care for critical patients with COVID-19, and (3) nurses who did not agree to participate in this study or refused to complete the interview. A total of 90 nurses worked in the ICU, and 85 nurses completed this study. The response rate was 94%. 

### 2.2. Quantitative Measurements

Questionnaires of quantitative measurements in this study included basic characteristics, the perceived stress scale (PSS-14), and the brief coping orientation to problems experienced inventory.

#### 2.2.1. Basic Characteristics

The basic characteristics included demographics and work-related characteristics. Demographics included age, gender, educational level (junior college, bachelor, or master), religious beliefs (no or yes), marital status (single or married), living status (living alone, living with friends, or living with family), and perceived friend or family support (1–5 points, 1 indicating no perceived support, and 5 indicating full perceived support from friends or family). Work-related characteristics included length of working experience in nursing, nursing level (levels 0–1 indicating low professional nursing level, and levels 2–4 indicating high professional nursing level), experience of participation in a major disaster, experience of participation in infection control and PPE education, experience of participation in stress-related education, and perceived preparedness for COVID-19 (1–5 points, 1 indicating no perceived preparedness, and 5 indicating full perceived preparedness).

#### 2.2.2. The Perceived Stress Scale (PSS-14)

We used the Taiwanese version of the perceived stress scale developed by Cohen et al. [15], which was translated by Chen [16]. It is a self-reported scale used to measure the general perceived stress level during the past month. It contains 14 items (7 negative items and 7 positive items), with the score ranging from 0 to 4 where 0 represents “never” and 4 represents “always”. The positive items were converted into negative items, and the total score range was 0–56 points. The higher the total score, the higher the perceived stress level. The total score was also categorized into low, moderate, and high perceived stress with ranges of 0–18 points, 19–37 points, and 38–56 points, respectively. The scale has been shown to have good reliability and validity in many populations. In this study, we recruited 85 nurses and measured their perceived stress levels. The reliability of Cronbach’s α was 0.669.

#### 2.2.3. Brief Coping Orientation to Problems Experienced (B-COPE) Inventory

The original coping orientation to problems experienced (COPE) inventory was developed by Carver et al. [17] using 60 items to assess the different ways in which people respond to stress. The brief coping orientation to problems experienced inventory (B-COPE) used in this study was revised by Carver [18]. The B-COPE has 28 items with distinct aspects of problem-focused coping behaviors, emotion-focused coping behaviors, and ineffective coping behaviors, which were divided into 14 subscales with 2 items in each subscale. Problem-focused coping behaviors included three subscales: active coping, planning, and use of instrumental support. Emotion-focused coping behaviors included five subscales: positive reframing, acceptance, humor, religion, and use of emotional support. Ineffective coping behaviors included six subscales: self-distraction, denial, venting, substance use, behavioral disengagement, and self-blame. Each item was rated on a Likert scale ranging from 1 to 4, with 1 representing “I haven’t been doing this at all” and 4 representing “I’ve been doing this a lot”. Each subscale ranged from 2 to 8, with a higher score indicating a higher frequency of coping behaviors with stress. The reliability of Cronbach’s α in these subscales was from 0.50 to 0.90 [18]. In this study, the reliability of Cronbach’s α was 0.736 for 85 nurses. 

### 2.3. Qualitative Data Collection

A qualitative data guide was developed using semi-structured questions. The participants were encouraged to write freely about their experience of stress and coping behavior during the period of caring for critically ill patients with COVID-19. The interview questions were: (1) What are your main stressors in caring for critical patients with COVID-19? (2) Which factors aggravate the stress? (3) How do you manage stress while caring for critically ill patients with COVID-19? (4) Could you describe how you expect the stress to be reduced?

### 2.4. Study Procedure

This study was approved by the Institutional Review Board (IRB) of our medical center (IRB No: B202005010). Participants who cared for critical patients with COVID-19 in this special ICU set up expeditiously and who fulfilled the inclusion criteria were recruited. The researcher explained the objectives of this study and the study protocol in a private meeting room at the medical center. After obtaining informed consent from each participant, the researcher collected data—both qualitative and quantitative, via questionnaires. The researcher repeatedly conducted the study procedure to collect data within two weeks. The collected information was considered confidential. Participants spent 20–30 min completing the questionnaires and were informed that they could withdraw from the study at any time without giving any reason.

### 2.5. Statistical Analysis

The quantitative data were analyzed using SPSS software (version 22.0; IBM Corp., Armonk, NY, USA) for Windows. Continuous variables were described as mean and standard deviations (SD), and categorical variables were described as frequencies and proportions. Multiple linear regression was used to analyze the predictors of perceived stress among critical care nurses participating in COVID-19 care. The statistical significance was set at *p* < 0.05. Qualitative data were analyzed using content analysis according to Graneheim and Lundman [19]. To grasp the responses thoroughly, the qualitative data were read several times. Meaning units that appropriated the objectives were marked and consolidated into shorter passages. They were then counted together in codes and classified into categories and subcategories to summarize the text further. Finally, a theme emerged as a result of qualitative data analysis. 

## 3. Results

### 3.1. Results of Quantitative Data

#### 3.1.1. Basic Characteristics of Participants

The mean age of the nurses was 30.6 years. Most critical care nurses were female (89.4%), had a bachelor’s degree (87.1%), had no religious beliefs (57.6%), were single (81.2%), and lived with friends (55.3%). The mean score of perceived friend or family support was 4.1 points. The mean length of working experience in nursing was 6.1 years. Most of the critical care nurses were at nursing level 2–4 and had no experience of participation in major disaster response (58.8%), infection control, PPE education (65.9%), or stress-related education (84.7%). The mean score of perceived preparedness for COVID-19 was 3.4 points (Table 1).

#### 3.1.2. Perceived Stress and Coping Behavior

The mean score of perceived stress level was 25.4 (SD = 6.2), and most nurses were perceived to have moderate stress (85.9%). Among the 14 subscales of the B-COPE, active coping, planning, and acceptance were rated as the most practiced coping behaviors followed by instrumental support, positive reframing, and venting. The lowest rated subscales were substance use, denial, and behavioral disengagement (Table 2). 

#### 3.1.3. Predictors of Perceived Stress

As shown in Table 3, perceived support from friends or family members, and length of working experience in nursing were significant predictors of perceived stress among the nurses after adjusting for educational level, living status, perceived friend or family support, length of working experience in nursing, and perceived preparedness. Participants with lower mean scores for perceived friend or family support (β = −1.6, 95% CI = −3.1–−0.1, *p* = 0.046), and shorter lengths of working experience in nursing (β = −0.4, 95% CI = −0.5–−0.1, *p* = 0.021) had higher perceived stress scores. 

### 3.2. Results of Qualitative Data

From the results of the quantitative data, we determined the perceived stress, coping behavior level, as well as the predictors of perceived stress among critical care nurses. However, we wanted to understand the details of the different kinds of stress and the ways to cope in this population. Therefore, we used qualitative questions to explore the information in detail. The results of the qualitative data were classified into themes for perceived stress and included “fear and worry” and “increased burden”. The themes emerged from three categories: “fear of infecting others”, “uneasy situations” and “inadequate nursing staffs”. The themes for coping behavior were “rest and support” and “affirmative fighting”. The themes emerged from the four categories: “seeking support”, “moderate relaxation”, “following protection procedures” and “positive acceptance”. An overview of the results, including themes, categories, and subcategories, is shown in Table 4.

#### 3.2.1. Perceived Stress

##### Fear and Worry

The theme “fear and worry” contained experiences of critical care nurses’ perceived stress while caring for patients with COVID-19. In the interviews, the nurses expressed that they were afraid of getting infected with COVID-19, they were fearful of infecting others, and were worried about insufficient protection. They also found themselves in uneasy situations where they worried about inadequate competency to care for critical patients with COVID-19, environmental changes, and patients’ rapidly changing conditions. 

##### Fear of Infecting

Critical care nurses described perceived stress while caring for critically ill patients with COVID-19 and their fear of getting infected. The subcategory “fear of being infected” contained descriptions of critical care nurses caring for patients with COVID-19. This was exemplified by statements such as, “This was a temporary micro-negative pressure unit, unlike the negative pressure unit, which has a safe negative pressure design. Therefore, when caring for patients with COVID-19, there are doubts about being infected by patients”. 

In the subcategory of “fear of infecting others”, critical care nurses were afraid of transferring COVID-19 to their family, friends, and relatives. One nurse stated, “I feel that this is a high-risk job, and I am afraid of infecting my family. Therefore, I do not go home. Moreover, there are a lot of people diagnosed with COVID-19 every day. I feel that the COVID-19 pandemic will never end”. 

In the subcategory of “worry about insufficient protection”, it was expressed that critical care nurses do not have adequate PPE and were exposed to an unsafe environment. One nurse stated, “The PPE is not enough, and the PPE is not easy to use. The PPE does not cover my feet and I often feel that it will be stained when I take it off”.

##### Uneasy Situations

In general, critical care nurses experienced uneasy situations while caring for patients with COVID-19. The subcategory of “worry about inadequate competency” explained how critical care nurses liked to stay an uneasy situation as told by one nurse, “It used to be a surgical intensive care unit, which used to take care of critically ill surgical patients. I have no experience in caring for critical patients in the infection unit. I was worried that the training time was too short, so I was not able to take care of patients with COVID-19”. 

While setting up the special ICU to care for critically ill patients with COVID-19, critical care nurses were worried about environmental changes. One nurse stated that “It used to be a surgical ICU, but it was reopened as a special ICU temporarily. Currently, it is an infective medical ICU. There are many things that need to be communicated with the medical care team”. 

In the subcategory “patient’s condition changed rapidly”, critical care nurses stated that “the vital signs of critical patients with COVID-19 changed dramatically, and they were unable to deal with the patient’s critical conditions in time”.

##### Increasing Burden

The theme “increasing burden” was due to an inadequate number of nurses because of long-term work and where the ratio of nurse-to-patient did not decrease. 

##### Inadequate Nurse Staffing

When caring for patients with COVID-19, critical care nurses required more time to put on their PPE, perform nursing interventions, and required other nurses to help them care for patients, such as changing patients’ positions. Nurses also stated that it took a lot of time to put on their PPE, and therefore, they did not change their PPE unnecessarily.

The “long-term work” subcategory was exemplified by this statement, “During the COVID-19 pandemic, nurses in this special ICU were diverted and our vacations were too short”.

In the subcategory of “nurse-to-patient ratio has not decreased”, it was shown that it caused a burden on staff due to unfamiliar routines. The nurse-to-patient ratio was higher. Several critical nurses stated, “We hoped that the hospital would offer enough nurses to care for these critically ill patients with COVID-19 to avoid nurse burnout”.

#### 3.2.2. Coping Behavior

##### Rest and Support

The theme “rest and support” contained experiences of coping behavior by critical care nurses while caring for patients with COVID-19. Nurses sought support from relatives and friends, substantial bonuses, and vocational feedback. Their coping strategies included moderate relaxation through leisure and entertainment, moderate rest, adequate sleep, enjoying food, and shopping.

##### Seeking Support

Critical care nurses usually sought support from relatives and friends to cope with stressful events during the COVID-19 pandemic. Within the subcategory “seeking support from relatives and friends”, critical care nurses explained their coping strategies. “To speak to my family and friends was a way to release my stress” stated one nurse. In the subcategory “substantial bonuses and vocational feedback”, most nurses expressed that if they could get extra bonuses soon, their stress would be diminished. 

##### Moderate Relaxation

Critical care nurses faced considerable stress when caring for patients with COVID-19. When describing how they coped with stress, most nurses mentioned that they needed leisure and entertainment, moderate rest, adequate sleep, and to enjoy food and shopping. Within the subcategory of “leisure and entertainment”, most critical care nurses stated they would exercise, listen to music, play games on the internet, watch movies, and read books. Within the subcategory “moderate rest and adequate sleep”, nurses expressed that they would sleep much more after work to decrease their stress. Within the subcategory “enjoy food and shopping”, nurses described that they liked to eat and drink after finishing work and found that their stress decreased instantly.

##### Affirmative Fighting

The theme “affirmative fighting” means that critical care nurses need to have a positive attitude to help them face the challenges of stress from caring for patients with COVID-19. It was found that nurses followed protection procedures and used positive acceptance to achieve “affirmative fighting”.

##### Following Protection Procedures

Critical care nurses protected themselves using infection control measures. Within the subcategory “following infection control regulations”, critical care nurses explained how they coped with their stress and approximately half of the nurses stated that they took care of themselves by complying with the COVID-19 prevention regulations to cope with stress. In the subcategory “appropriate protective equipment”, most nurses expressed that they wore the PPE, and properly performed the dressing and undressing procedure to cope with stress.

##### Positive Acceptance

In general, most nurses displayed a positive acceptance while dealing with stress. They also hoped that hospital management would ensure a friendly working environment. Within the subcategory “positive thinking”, critical care nurses stated that they had to find a way to solve their problems to survive the COVID-19 crisis. In the subcategory “friendly environment”, critical care nurses stated that their hospitals needed to provide more supplies to nurses and supply vaccines to nurses caring for patients with COVID-19.

## 4. Discussion

The finding of this study showed that the longer the working experience, the lower the perceived stress among critical care nurses caring for critically ill COVID-19 patients. This finding is inconsistent with the results of previous studies [20,21], but consistent with the prior study [22]. Nurses with less working experience usually faced many challenges, such as increasing workload, a lack of professional autonomy, high leader role expectations, and role ambiguity leading to conflict [23]. These factors all contribute to the perceived stress of critical care nurses.

Our study results also showed that the higher the perceived support from friends or family members, the lower the perceived stress among critical care nurses caring for critically ill COVID-19 patients. One of our qualitative results for the category of coping behavior was “seeking support”. Most nurses described how they sought support from their relatives and friends. This finding is similar to those of several previous studies [7,10,24]. Critical care nurses caring for COVID-19 patients may experience emotional and physical symptoms, care environment challenges, and social effects. Therefore, their coping strategies included support from co-workers and family, distractions, mind/body wellness, and spirituality/faith [25]. Chinese individuals tend to be close to their families or friends, who consequently end up becoming the most important sources of emotional support in Chinese culture. Family and friend support can provide relief to nurses from stress and reduce their worries [7,24]. In addition to seeking support from relatives and friends, most nurses described how they needed substantial bonuses and vocational feedback. Prior studies also indicated that financial appreciation was a key factor in reducing stress during the COVID-19 outbreak [8,10]. In another “moderate relaxation” category of the theme “rest and support”, critical care nurses expressed that suitable leisure and entertainment, moderate rest, adequate sleep, shopping, and enjoying food reduced their stress. A prior study indicated that sharing jokes with colleagues [8] and engaging in leisurely activities also helped deal with stress [24].

The top three coping behaviors practiced by the nurses in this study were active coping, planning, and acceptance. These quantitative results were consistent with the qualitative results of the theme “affirmative fighting” that emerged from “following protection procedures” and “positive acceptance”. In the category ‘following protection procedures’, critical care nurses stated that following infection control regulations and using appropriate PPE reduced their stress. In the category “positive acceptance”, critical care nurses thought that positive thinking and a friendly working environment reduced stress while caring for patients with COVID-19. These findings are similar to those of prior studies [8,10,24]. Indeed, nursing colleagues who had a positive attitude and clear guidelines from the hospital regarding infection prevention reduced stress during the COVID-19 outbreak [8,10,24].

In the qualitative results, the theme of perceived stress “fear and worry” emerged from the two categories “fear of infecting” and “uneasy situations”. The other theme of perceived stress, “increased burden”, emerged from the category “inadequate nursing staffs” Almost all nurses described their fears of being infected and infecting others, and worried about insufficient protection while caring for critical patients with COVID-19. A prior study indicated that various factors caused stress among health workers, including wearing PPE every day, a perception of insufficient protective measures against COVID-19, and worrying about transmitting the virus to their family or friends [10]. Another study showed that healthcare workers were anxious about their safety, and the safety of their families [24]. Another study surveyed 1208 healthcare workers during the COVID-19 outbreak in Wuhan. The results showed that although the healthcare workers worried about getting infected and infecting family members, and found working while wearing PPE uncomfortable, they continued providing care to COVID-19 patients out of a sense of professional integrity [5], abiding by professional ethics and obligations, and valued their jobs [10,26]. Nurses felt more nervous and scared than other health care members because they were directly involved in taking care of patients during the COVID-19 outbreak [8]. Critical care nurses also explained that they worked in difficult circumstances due to worry about inadequate competency, environmental changes, and the rapid decline in health of their patients. A prior study indicated that nurses were stressed not knowing when the COVID-19 pandemic would be under control and witnessing patients die from COVID-19 [8]. Another study showed that nurses who were younger and had less work experience felt inadequate and had higher levels of stress while caring for patients with COVID-19 [27]. In the category of “inadequate nurses”, critical care nurses complained about long-term work, and inadequate nurse-to-patient ratios. A prior study showed that nurses were stressed because they faced extremely demanding workloads and a shortage of staff at times during the COVID-19 outbreak [8]. Not knowing when the pandemic would come to an end was recognized as a significant stress for many health professionals in addition to caring for patients with COVID-19. Importantly, nursing staff usually reported feelings of stress from psychological exhaustion and fatigue due to long-term work, or inadequate nurse staffing to care for the patients.

This study has several strengths and limitations. This was a mixed-method study, and almost all nurses from this special ICU caring for critically ill patients with COVID-19 completed the study. The results of this study can be generalized to nurses caring for critically ill patients with COVID-19 in this medical center. However, due to the COVID-19 pandemic, we collected qualitative data using open questions that they answered themselves in writing, rather than face-to-face interviews. Therefore, the qualitative data may be limited. It was challenging to understand the nurses’ emotions and facial expressions without seeing them in person. Furthermore, this study focused on a single experience of a medical center from May to July 2021, the peak of the outbreak of COVID-19, and therefore the generalizability was limited. We suggest investigating nursing experiences from more hospitals to confirm these findings.

## 5. Conclusions

The findings of this study showed that nurses who perceived less support from their friends or families had higher perceived stress. The major perceived stress for critical nurses came from the fear and worry of infecting others, being in uneasy situations, and the increased burden of a short supply of nursing care for critical patients with COVID-19. They used coping strategies such as rest, support, and affirmative fighting to deal with stress. Based on these findings, it is proposed that the support that nurses receive from their families is an important predictor of perceived stress. Therefore, it is suggested that nurses caring for critical patients with COVID-19 be provided with more support to decrease their stress.

## Figures and Tables

**Table 1 ijerph-19-04258-t001:** Basic characteristics of critical care nurses caring for critical patients with COVID-19 (*n* = 85).

Variable	Mean ± SD/*n* (%)
Demographics	
Age (years)	30.6 ± 6.9
Gender	
Male	9 (10.6)
Female	76 (89.4)
Educational level	
Junior college	5 (5.9)
Bachelor	74 (87.1)
Master	6 (7.1)
Religious Belief	
No	49 (57.6)
Yes	36 (42.4)
Marital Status	
Single	69 (81.2)
Married	16 (18.8)
Living Status	
Living alone	16 (18.8)
Living with friends	47 (55.3)
Living with family	22 (25.9)
Perceived friends or family support	4.1 ± 0.8
Work-related characteristics	
Length of service in nursing (years)	6.1 ± 6.1
Identity	
Military	22 (25.9)
Private employment	63 (74.1)
Nursing level	
Level 0~1	34 (40.0)
Level 2~4	51 (60.0)
Participated in major disaster	
No	50(58.8)
Yes	35 (41.2)
Participate in infection control and PPE education	
No	29 (34.1)
Yes	56 (65.9)
Stress-related education	
No	72 (84.7%)
Yes	13 (15.3%)
Perceived preparedness for COVID-19	3.4 ± 0.8

SD = Standardized deviation; PPE = personal protective equipment.

**Table 2 ijerph-19-04258-t002:** Perceived stress, and coping behavior among critical care nurses caring for critical patients with COVID-19 (*n* = 85).

Variable	Mean ± SD
Stress	25.4 ± 6.2
B-COPE	
Problem-focused coping behaviors	
Active coping	6.1 ± 1.5
Planning	6.1 ± 1.5
Use of instrumental support	6.0 ± 1.5
Emotion-focused coping behaviors	
Positive reframing	5.9 ± 1.4
Acceptance	6.1 ± 1.6
Humor	5.4 ± 1.5
Religion	4.4 ± 1.7
Use of emotional support	5.7 ± 1.5
Ineffective coping behaviors	
Self-distraction	5.7 ± 1.4
Denial	4.0 ± 1.5
Venting	5.8 ± 1.5
Substance use	3.6 ± 1.7
Behavioral disengagement	4.0 ± 1.4
Self-blame	5.1 ± 1.5

SD = Standardized deviation; B-COPE = Brief Coping Orientation to Problems Experienced Inventory.

**Table 3 ijerph-19-04258-t003:** Predictors of perceived stress among critical care nurses caring for critical patients with COVID-19 (*n* = 85).

Variable	Crude β (95% CI)	*p* Value	Adjusted β # (95% CI)	*p* Value
Demographics				
Age (years)	−0.2 (−0.4–0.0)	0.066	0.1 (−0.2–0.4)	0.585
Gender				
Male	Reference		Reference	
Female	2.8 (−1.4–7.0)	0.197	2.5 (−1.3–6.3)	0.195
Educational level				
Junior college	Reference		Reference	
Bachelor	−4.2 (−9.6–1.3)	0.140	−4.5 (−9.7–0.7)	0.093
Master	−8.3 (−15.4–−1.1)	0.026	−4.7 (−11.5–2.2)	0.184
Religious Belief				
No	Reference		Reference	
Yes	−0.4 (−3.0–2.3)	0.794	0.1 (−2.4–2.4)	0.998
Marital Status				
Single	Reference		Reference	
Married	−3.0 (−6.3–0.3)	0.075	0.4 (−3.5–4.3)	0.844
Living Status				
Living alone	Reference		Reference	
Living with friends	−1.6 (−5.1–1.8)	0.355	−0.3 (−3.4–2.8)	0.846
Living with family	−4.1 (−8.0–−0.2)	0.040	−2.1 (−5.8–1.7)	0.283
Perceived family support	−2.4 (−3.9–−0.9)	0.002	−1.6 (−3.1–−0.1)	0.046
Work-related characteristics				
Length of working experience in nursing (years)	−0.3 (−0.5–−0.1)	0.010	−0.4 (−0.5–0.1)	0.021
Nursing level				
Level 0~1	Reference		Reference	
Level 2~4	−2.0 (−4.7–0.6)	0.137	0.1 (−2.6–2.6)	0.998
Participated in major disaster				
No	Reference		Reference	
Yes	−1.5 (−4.1–1.2)	0.285	0.5 (−2.5–3.5)	0.760
Participate in infection control and PPE education				
No	Reference		Reference	
Yes	−2.6 (−5.4–0.1)	0.060	−1.8 (−4.4–0.8)	0.169
Stress-related education				
No	Reference		Reference	
Yes	−1.7 (−5.3–2.0)	0.376	0.5 (−3.1–4.1)	0.784
Perceived preparedness	−2.6 (−4.2–−0.9)	0.003	−1.7 (−3.3–0.0)	0.051

PPE = personal protective equipment. # All results of adjusted β were adjusted by educational level, living status, perceived friend or family support, length of working experience in nursing, and perceived preparedness.

**Table 4 ijerph-19-04258-t004:** The results of the qualitative data analysis presented as themes, categories, and subcategories.

Topic	Theme	Category	Subcategory
Perceived stress	Fear and worry	Fear of infecting	Fear of being infected
			Fear of infecting others
			Worry about insufficient protection
		Uneasy situation	Worry about inadequate competency
			Worry about environmental change
			Patient’s condition changed rapidly
	Increased burden	Inadequate staffing	Long-term work
			Nurse-to-patient ratio has not decreased
Coping behavior	Rest and support	Seeking support	Seeking support from relatives and friends
			Substantial bonuses and vocational feedback
		Moderate relaxation	Leisure and entertainment
			Moderate rest and adequate sleep
			Enjoy food and shopping
	Affirmative fighting	Following protection procedures	Following infection control regulations
			Appropriate protective equipment
		Positive acceptance	Positive thinking
			Friendly environment

## Data Availability

The data presented in this study are available on request from the corresponding author. The data are not publicly available due to privacy.

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
