# Peer review of "Perceived Stress and Coping Behavior of Nurses Caring for Critical Patients with COVID-19 Outbreak in Taiwan: A Mixed-Methods Study"

_ijerph, 2022, doi:10.3390/ijerph19074258_

Round 1
Reviewer 1 Report
In “Perceived Stress and Coping Behavior of Nurses Caring for Critical Patients with COVID-19 Outbreak in Taiwan: A Mixed-Methods Study” the authors explore the perceived stress and copying behaviours of nursing caring for critically ills patients with Covid-19. The manuscript is clear, relevant for the field and well written. The results seem plausible. I have just one aspect that I suggest for improving the manuscript, which is the addition of the inclusion and exclusion criteria in methodology.
I recommend to accept after minor revision.

Author Response
Reviewer1#
In general, the manuscript is clear, relevant for the field and well written. The results seem plausible and appropriate. Before accepting the manuscript, I would suggest minor revisions which I believe that could improve some aspects.
Authors’ response:
Thank you very much.
These improvements are the following:
- The addition of the inclusion and exclusion criteria in methodology. It would be important if the authors let this feature clear and even better if these criteria had been determined before making the interview. As it is well-known, particularly important for small samples, the exclusion of some specific interviewees may significantly alter the outcome of the research.
Authors’ response:
Thank you very much for the suggestions. We have revised and highlighted the inclusion and exclusion criteria in blue in the Study design and participants of Methods section and used the Track Changes.
- Although the authors say that ‘few works’ have explored similar topic as in this manuscript, it would benefit the manuscript, if current works could be added in similar line and more clear comparison with other works could be made.
Authors’ response:
We agree with the comments. We have revised to make more clear comparison with other works in the last paragraph of Introduction using the Track Changes.
- The results of qualitative analysis could be better explored. There is only one table for describing the qualitative result and much less discussion when compared with the quantitative analysis. A literature discussion in this topic would increase the quality of manuscript.
Authors’ response:
Thank you very much for your valuable suggestions. Although there was only one table for describing the qualitative results, we explained the details in “3.2 Results of qualitative data”. We also rechecked and revised the Discussion section and highlighted in blue using Track Changes.

Reviewer 2 Report
I have mixed feelings regarding this manuscript: "Perceived Stress and Coping Behavior of Nurses Caring for Critical Patients with COVID-19 Outbreak in Taiwan: A Mixed-Methods Study".
. I find the topic to be of overall interest and very important, yet I note several difficulties with the manuscript. There are strengths. The paper proposes an interesting exploration of the mental condition of nurses from a special intensive care unit (ICU) of a medical center. In my opinion, it is not enough.
Some of the most serious concerns about this manuscript are noted below.
Abstract:
- "The questionnaire results showed that the average perceived stress level among nurses was 25.4 points." What does it mean? It is not clear if it is a high or low score.
- Does the information in parenthesis shouldn't be at the end of the sentence? ("Nurses who received less perceived support from their families (β= 27 -1.7, 95% CI= -3.2 – -0.1, p = 0.040) were significantly affected by perceived stress")
- It is not clear: "The average coping behaviors of nurses were active coping, planning, and acceptance." Whether average is more common?
- Does the presentation of qualitative data at the end of the abstract ("The interviews revealed that the major causes of nurses' perceived stress came from fear, worry...") shouldn't come at the beginning, before statistical data.
Main text
- It would be better if the qualitative part of the study became the starting point for quantitative analyses. More advanced statistics should follow simpler analyses. This organization of text has no explanation. A lot part of the article is qualitative analyses. Two kinds of presented analyses are not compatible. The authors should try to combine results received from mixed methods.
- Moreover, logistic regression has many predictors, and the group is relatively small for all those variables. Researchers should once again conduct quantitative analyzes adjust the number of variables to the size of the sample. Perhaps try to use data from qualitative to quantitative analysis. Presentes statistics is elementary; it seems too simple.
- It seems that in the presented statistics, we have only one result from quantitative analyses ("nurses who perceived less support from their families had higher perceived stress"). This finding is not novel in the context of studies about social support.
- This kind of information (section Results and other places, also in the abstract) "The mean score of perceived stress level was 25.4 (SD=6.2)", "The mean score of perceived stress level was 25.4 (SD=6.2)" is redundant. It is not readable. We don't know the maximum and minimum or average scores in different studies.
- We can read in the article "Participants - were single (81.2%), and lived with friends (55.3%)" Is not support from friends more important than support from family, especially for those nurses who live with friends? This information has a meaning in the context of perceived support. With who live the second part of participants (about 45%)? With family? Why did the researcher verify only support from family?
- We don't know if the result can be generalized. The authors do not write about this in the article.
- It this: "The Institutional Review Board approved this study of our medical center 157 (B202005010), means approval of the committee of ethics?
Author Response
Abstract:
"The questionnaire results showed that the average perceived stress level among nurses was 25.4 points." What does it mean? It is not clear if it is a high or low score.
Authors’ response:
Thank you very much for the comments. “The total score was also categorized into low, moderate, and high perceived stress with ranges of 0-18 points, 19-37 points, and 38-56 points, respectively.” We have added the above content in 2.2.2 The perceived stress scale. In addition, we have revised the results in Abstract and highlighted in blue using Track change.
Does the information in parenthesis shouldn't be at the end of the sentence? ("Nurses who received less perceived support from their families (β= 27 -1.7, 95% CI= -3.2 – -0.1, p = 0.040) were significantly affected by perceived stress")
Authors’ response:
We agree with the suggestions. We have revised the sentence in Abstract and highlighted in blue as well as used Track Changes.
It is not clear: "The average coping behaviors of nurses were active coping, planning, and acceptance." Whether average is more common?
Authors’ response:
Thank you very much for the comments. We have revised the sentence in Abstract and highlighted in blue as well as used Track Changes.
Does the presentation of qualitative data at the end of the abstract ("The interviews revealed that the major causes of nurses' perceived stress came from fear, worry...") shouldn't come at the beginning, before statistical data.
Authors’ response:
Thank you very much for the comments. From the results of the quantitative data, we determined the perceived stress, coping behavior level, as well as the predictors of perceived stress among critical care nurses. However, we wanted to understand the details of the different kinds of stress and the ways to cope in this population. We used qualitative questions to explore the information in details. Therefore, we wrote the quantitative results first, and then wrote the qualitative results.
Main text
- It would be better if the qualitative part of the study became the starting point for quantitative analyses. More advanced statistics should follow simpler analyses. This organization of text has no explanation. A lot part of the article is qualitative analyses. Two kinds of presented analyses are not compatible. The authors should try to combine results received from mixed methods.
Authors’ response:
We agree with the valuable comments. We knew the perceived stress, and the coping behaviors of critical care nurses, as well as the predictors of perceived stress from the results of quantitative data. We used qualitative questions to explore the details about what kinds of stress and how to cope in this population. We have tried to combine results in Discussion section using Track changes.
- Moreover, logistic regression has many predictors, and the group is relatively small for all those variables. Researchers should once again conduct quantitative analyzes adjust the number of variables to the size of the sample. Perhaps try to use data from qualitative to quantitative analysis. Presentes statistics is elementary; it seems too simple.
Authors’ response:
We agree with the comments. We used univariate analysis to examine the correlation between perceived stress and demographics and work-related characteristics. Then, the significant factors in univariate analysis were used to adjust in multivariate regression analysis to analyze the predictors of perceived stress. The details were shown in Table 3.
- It seems that in the presented statistics, we have only one result from quantitative analyses ("nurses who perceived less support from their families had higher perceived stress"). This finding is not novel in the context of studies about social support.
Authors’ response:
Thank you very much for the comments. From the results of the quantitative data, we determined the perceived stress, coping behavior level, as well as the predictors of perceived stress among critical care nurses. However, we wanted to understand the details of the different kinds of stress and the ways to cope in this population. Therefore, we used qualitative questions to explore the information in detail.
- This kind of information (section Results and other places, also in the abstract) "The mean score of perceived stress level was 25.4 (SD=6.2)", "The mean score of perceived stress level was 25.4 (SD=6.2)" is redundant. It is not readable. We don't know the maximum and minimum or average scores in different studies.
Authors’ response:
We agree with the comments. “The results showed that the average perceived stress level among nurses was 25.4 points, and most of them perceived moderate stress.” We have revised the information in the Abstract and Results sections.
- We can read in the article "Participants - were single (81.2%), and lived with friends (55.3%)" Is not support from friends more important than support from family, especially for those nurses who live with friends? This information has a meaning in the context of perceived support. With who live the second part of participants (about 45%)? With family? Why did the researcher verify only support from family?
Authors’ response:
We agree with the reviewer’s suggestions that the information has a meaning in the context of perceived support from friends. We found that nurses perceived support from their friends and their families were the same. Therefore, we combine the two. We have revised in the manuscript and Table. Thank you.
- We don't know if the result can be generalized. The authors do not write about this in the article.
Authors’ response:
“This was a mixed-method study, and almost all nurses from this special ICU caring for critically ill patients with COVID-19 completed the study. The results of this study can be generalized to nurses caring for critically ill patients with COVID-19 in this medical center.” We have revised in the strengths and limitations. Thank you.
- Is this: "The Institutional Review Board approved this study of our medical center 157 (B202005010), means approval of the committee of ethics?
Authors’ response:
Yes, " This study was approved by the Institutional Review Board (IRB) of our medical center (IRB No: B202005010)”, means approval of the committee of ethics.

Reviewer 3 Report
MINOR REVISION
In the discussion, page 7, lines 330-331, the authors can improve the framing of the discussion. The study cited did not find a significant correlation between a maladaptive coping category and the 3 burnout dimensions. The authors statement "That may be because that religion was the most important in Islam’s life " is an overreach in what aspects of the study result the authors are comparing. In addition, needs correct use of English language.
Author Response
Reviewer3#
In the discussion, page 7, lines 330-331, the authors can improve the framing of the discussion. The study cited did not find a significant correlation between a maladaptive coping category and the 3 burnout dimensions. The authors statement "That may be because that religion was the most important in Islam’s life " is an overreach in what aspects of the study result the authors are comparing. In addition, needs correct use of English language.
Authors’ response:
Thank you very much for the valuable comments. We have revised and rechecked the Discussion section, highlighted in blue and corrected use of English language by English expert.
